# Electron Beam Radiation as a Safe Method for the Sterilization of Aceclofenac and Diclofenac—The Usefulness of EPR and ^1^H-NMR Methods in Determination of Molecular Structure and Dynamics

**DOI:** 10.3390/pharmaceutics14071331

**Published:** 2022-06-24

**Authors:** Marcin Janiaczyk, Anna Jelińska, Aneta Woźniak-Braszak, Paweł Bilski, Maria Popielarz-Brzezińska, Magdalena Wachowiak, Mikołaj Baranowski, Szymon Tomczak, Magdalena Ogrodowczyk

**Affiliations:** 1Department of Pharmaceutical Chemistry, Poznan University of Medical Sciences, Grunwaldzka 6, 60-780 Poznan, Poland; m.janiaczyk@ziololek.com.pl (M.J.); ajelinsk@ump.edu.pl (A.J.); mpopiel@ump.edu.pl (M.P.-B.); szymon.tomczak@ump.edu.pl (S.T.); 2Pharmaceutical Company “Ziołolek” Sp. z o.o., Starolecka 189, 61-341 Poznan, Poland; 3Functional Materials Physics Division, Faculty of Physics, Adam Mickiewicz University, Uniwersytetu Poznańskiego 2, 61-614 Poznan, Poland; awbraszak@gmail.com (A.W.-B.); zfwc@amu.edu.pl (M.W.); mikbar@amu.edu.pl (M.B.); 4Medical Physics and Radiospectroscopy Division, Faculty of Physics, Adam Mickiewicz University, Uniwersytetu Poznańskiego 2, 61-614 Poznan, Poland; bilski@amu.edu.pl

**Keywords:** ionizing radiation, diclofenac, aceclofenak, FT-IR, HPLC, ^1^H-NMR, EPR

## Abstract

Diclofenac (DC) [2-(2,6-Dichloroanilino)phenyl]acetic acid,) and aceclofenac (AC) 2-[2-[2-[(2,6-dichlorophenyl)amino]phenyl]acetyl]oxyacetic acid in substantia were subjected to ionizing radiation in the form of a beam of high-energy electrons from an accelerator in a standard sterilization dose of 25 kGy and higher radiation doses (50–400 kGy). We characterized non-irradiated and irradiated samples of DC and AC by using the following methods: organoleptic analysis (color, form), spectroscopic (IR, NMR, EPR), chromatographic (HPLC), and others (microscopic analysis, capillary melting point measurement, differential scanning calorimetry (DSC)). It was found that a absorbed dose of 50 kGy causes a change in the color of AC and DC from white to cream-like, which deepens with increasing radiation dose. No significant changes in the FT-IR spectra were observed, while no additional peaks were observed in the chromatograms, indicating emerging radio-degradation products (25 kGy). The melting point determined by the capillary method was 153.0 °C for AC and 291.0 °C for DC. After irradiation with the dose of 25 kGy for AC, it did not change, for DC it decreased by 0.5 °C, while for the dose of 400 kGy it was 151.0 °C and 286.0 °C for AC and DC, respectively. Both NSAIDs exhibit high radiation stability for typical sterilization doses of 25–50 kGy and are likely to be sterilized with radiation at a dose of 25 kGy. The influence of irradiation on changes in molecular dynamics and structure has been observed by ^1^H-NMR and EPR studies. This study aimed to determine the radiation stability of DC and AC by spectrophotometric, thermal and chromatographic methods. A standard dose of irradiation (25 kGy) was used to confirm the possibility of using this dose to obtain a sterile form of both NSAIDs. Higher doses of radiation (50–400 kGy) have been performed to explain the changes in DC and AC after sterilization.

## 1. Introduction

Diclofenac sodium (DC), a phenylacetic acid derivative, is the most widely used non-steroidal anti-inflammatory drug (NSAID) globally. It was the product of rational drug design based on the structures of phenylbutazone, mefenamic acid, and indomethacin [1].

The addition of two chlorine groups in the ortho position of the phenyl ring locks the ring in maximal torsion, which appears to be related to increased potency. It demonstrates anti-inflammatory, analgesic, and antipyretic properties. DC possesses three mechanisms of action: inhibition of the arachidonic acid COX system, resulting in a decreased production of prostaglandins and thromboxanes and inhibition of the lipoxygenase pathway, resulting in decreased production of leukotrienes, as well as inhibition of arachidonic acid release and stimulation of its reuptake, resulting in a reduction of arachidonic acid availability. DC is rapidly and completely absorbed in oral administration. Only 50% to 60% of an oral dose is bioavailable due to extensive hepatic metabolism. Based on this process, four major metabolites resulting from aromatic hydroxylation have been identified—4′-hydroxy derivative, 5-hydroxy, 3′-hydroxy, and 4′,5-dihydroxy metabolites [2,3,4]. Aceclofenac (AC) was developed as an analog of diclofenac, via chemical modification to improve the drug’s gastrointestinal tolerability. AC potently inhibits the cyclooxygenase enzyme (COX) involved in synthesizing prostaglandins, which are inflammatory mediators that cause pain, swelling, inflammation, and fever. It displays high permeability to penetrate synovial joints. In patients with osteoarthritis and related conditions, the loss of articular cartilage causes joint pain, tenderness, stiffness, crepitus, and local inflammation [5]. AC is also effective in other painful states such as dental and gynecological conditions [6]. DC and AC are indicated to relieve pain and inflammation in osteoarthritis, rheumatoid arthritis, and ankylosing spondylitis. The indications for using DC and AC in ophthalmology include preventing and treating inflammation and cystic macular edema after cataract surgery, inhibition of intraoperative myosis during cataract surgery, and treatment of pain and allergic conjunctivitis [7,8]. DC is also used to treat migraine. DC and AC are used in the form of film-coated and dissolving tablets with prolonged action. DC is also available in the form of a gel, spray, and patches for topical treatment and suppositories, and a solution for injection [9,10]. Eye drops and parenteral forms of the drug require sterility. One of the methods of sterilization permitted by the European Pharmacopoeia [11] is radiation, which is a promising and an alternative method for obtaining a sterile form of drugs. Considering the rapid technological development and the introduction of newer forms of the drug, including polymer hydrogel and carbon nanotube for prolonged percutaneous release of DC [12], we performed the influence of ionizing radiation on the physico-chemical properties of DC and AC. Radiation stability of DC was previously discussed and concerned with the irradiation of new forms of drug delivery as lipid nanoparticles to the target/site of action [13]. For complete sterilization, doses below 25 kGy are sufficient, which do not change DC or the drug form. The other publication concerns using gamma radiation for the degradation of DC as an environmental pollutant in aqueous solutions [14] DC undergoes degradation according to the pseudo-first-order reaction, even at a dose below 1 kGy. In determining the stability of DC over the pH range of 5-8 and its inclusion complex with β-cyclodextrin in solution isothermal microcalorimetric technique was used. The degradation of DC and its inclusion complex was also a pseudo-first-order reaction [15]. Ionizing radiation shows bactericidal properties but can also cause physicochemical changes in sterilized drugs, as discussed in many publications [16,17,18,19]. Moreover, it was demonstrated that ionizing radiation in the standard dose of 25 kGy did not reduce the antimicrobial activity of many tested antibiotics [20,21,22]. Our research aims to explain how DC and AC behave in a solid state when exposed to radiation in the form of an electron beam and to assess their radiation stability.

## 2. Materials and Methods

### 2.1. Material

Diclofenac (DC); [2-(2,6-Dichloroanilino)phenyl]acetic acid; molecular formula C_14_H_10_C_l2_NNaO_2_, molecular weight 318,14, CAS number 15307-79-6, POL-AURA, Lot 009PFF.

Aceclofenac (AC); 2-[2-[2-[(2,6-Dichlorophenyl)amino]phenyl]acetyl]oxyacetic acid; molecular formula C_16_H_13_C_l2_NO_4_, molecular weight 354,18, CAS number 89796-99-6, POL-AURA, Lot 348LTV (Figure 1).

All other chemicals and solvents were supplied by Merck KGaA (Darmstadt, Germany) and were of analytical grade. High quality pure water was prepared using the Millipore purification system (Millipore, Molsheim, France, model Exil SA 67120).

### 2.2. Exposure to Irradiation

Approximately 0.1 g of each substance was placed in colorless 5 mL glass vials and closed with a plastic stopper. The vials were then exposed to radiation from a linear electron accelerator LAE 13/9 (electron beam 9.96 MeV and current intensity 6.2 A) until they absorbed the doses of 25 (standard dose [23]), 50, 100, 200, and 400 kGy.

### 2.3. Organoleptic Analysis

Before and after irradiation, the compounds were subjected to organoleptic analysis comparing their color (against a white background), form, odor, solubility, and clarity of the solutions to those of the non-irradiated samples.

### 2.4. Color Determination

The color of the non-irradiated and irradiated substances was carried out using NH310 Colorimeter device. Before the measurement, the device was subjected to white and black colors. The results were an average of three measurements.

### 2.5. Microscopic Analysis

Microscopic analysis was performed using Levenhuk 870T microscope equipped with Microscope Digital Camera M800 Plus Levenhuk (a 40× magnification lens and a 10× magnification eyepiece).

### 2.6. Melting Point Determination

The melting points of DC and AC were determined by the capillary method using MP 70 Melting Point System (Mettler Toledo, Greifensee, Switzerland). The measurements were performed for non-irradiated and irradiated samples (0, 25, 50, 100, 200, and 400 kGy). The samples of AC were heated from 25 °C to 153 °C, in the range of 135–153 °C, with a heating rate of 1 °C/min. The samples of DC were heated from 100 °C to 290 °C, in the range of 275–290 °C, with a heating rate 1 °C/min. The results were an average of three measurements.

### 2.7. Differential Scanning Calorimetry (DSC)

Differential scanning calorimetry thermograms were recorded using DSC 214 Polyma Netzsch (Netzsch Group, Selb, Germany) in a nitrogen atmosphere (30 mL/min). The samples were heated up to 250 °C with a scanning rate of 5 °C/min. The samples of approximately 5.0 mg of non-irradiated and irradiated substances were sealed in aluminum cells with pierced lids. The results were processed using the TAA (Netzsch) software. The peak area was evaluated using linear baseline corrections.

### 2.8. Electron Paramagnetic Resonance (EPR) Spectroscopy

EPR measurements were made on an Adani X-band spectrometer using the eSpinoza control software. The samples were prepared in the same way, each of them was weighed on a laboratory balance scale and placed in thin-walled glass vials with a diameter of 5 mm. The weights of the individual samples were correspondingly: DC 0.024 g, DC 25 kGy 0.014 g, AC 0.019, AC 25 kGy 0.007 g. During the measurements on the EPR spectrometer, the samples were always placed in the resonator in the same way and the tuning was performed to achieve the best possible detection condition. In order to obtain the possibility of comparing individual spectra from the side of the cavity, a standard TCNQ sample characterized by a very narrow line was placed in the resonator. The placed reference was secured against the possibility of changing the position so that it is always in the same place. Thanks to this, it was not necessary to take into account the quality factor of the resonance cavity in the analysis. During the analysis, scaling of individual spectra was performed in order to obtain the same amplitudes of the reference spectrum. The reference spectrum was then subtracted.

### 2.9. Fourier Transform Infrared Spectroscopy (FTIR)

The FTIR analyses were performed on Bruker FTIR IFS 66/s spectrometer (Billerica, MA, USA) using the KBr disc technique in the wave number range of 400–4000 cm^−1^ with a resolution of 1 cm^−1^. Discs of 15 mm diameter were prepared by mixing 1 mg of the substance to be studied with 300 mg of potassium bromide until a homogeneous mixture was obtained. Pay Unicam minipress was used to form the discs. Pure KBr disc was used as a blank sample. For each spectrum, 30 scans were taken.

### 2.10. Proton Nuclear Magnetic Resonance (^1^H-NMR)

The 1H NMR experiments were performed on a pulse spectrometer operating at 25 MHz (El-Lab Tel-Atomic, Poznań, Poland) [24,25,26,27]. The non-irradiated and irradiated samples of DC and AC were sealed in glass tubes and degassed to remove humidity effects and paramagnetic oxygen. Measurements were performed in a wide temperature range from 80 K to 300 K. The sample temperature was stabilized within 15 min and controlled using a gas-flow cryostat, and monitored with a Pt resistor to an accuracy of 0.5 K.

### 2.11. High-Performance Liquid Chromatography (HPLC)

The HPLC analysis was performed on an Agilent Technologies 1220 Infinity LC chromatograph (Santa Clara, CA, USA) with a binary pump, a DAD detector, autosampler, and a column oven. The mobile phase consisted of acetonitrile HPLC grade (phase A) and 0.1% acetic acid (pH of 2.8) (phase B). Phases A and B were mixed in a ratio of 50:50 (*v/v*), the flow rate was 1.0 mL/min, and the injection volume was 20 μL. The stationary phase was C18 silica gel for chromatography (Xterra C18 column; 250 mm × 4.6 mm, 3.5 µm). The analyses were performed at 30 ± 1 °C with monitoring the signal at 216 nm. The run-time of the experiment was 15 min. The method was validated according to the Guidelines of the International Conference on Harmonization [28]. The content of DC and AC was calculated by reference to the standard. The difference in the content between the irradiated DC and AC and the calculated content of DC and AC brought information on the loss of the active substance as a result of irradiation.

#### 2.11.1. Method Validation

##### Linearity

The linearity was evaluated using measurements of six concentrations of the analyte. The concentrations ranged from 0.5038 × 10^−4^ to 6.0456 × 10^−4^ g/mL and from 0.5014 × 10^−4^ to 6.0168 × 10^−4^ g/mL for AC and DC, respectively. Three separate series of calibration standards were prepared to establish linearity. The results were examined for a linear relationship by plotting the analyte Pi versus the corresponding concentrations, followed by ordinary least squares linear regression (OLS) and calculating the slope, intercept, and correlation coefficient.

##### Precision and Repeatability

The repeatability (intra-day precision) and intermediate precision (inter-day precision) were calculated by analyzing nine AC and DC samples at 2.66 × 10^−4^ g/mL concentrations. The results were expressed as the relative standard deviation (RSD). The intra-day precision was performed on the same day, and another analyst repeated the inter-day precision the next day.

##### Accuracy

The accuracy was performed to obtain the closeness of the agreement between the expected value and the determined value. Nine different samples of DC and AC at a concentration of 2.66 × 10^−4^ g/mL were analyzed.

##### Limit of Determination (LOD) and Limit of Quantification (LOQ)

LOD was calculated as 3.3 Sy/a, and LOQ as 10 Sy/a, where Sy is a standard error and a is the slope of the corresponding calibration curve

## 3. Results and Discussion

Diclofenac (DC) and aceclofenac (AC) were white, crystalline, odorless compounds. The color of DC and AC was reported according to the CIEL*a*b* system (the color space defined by the International Commission on Illumination), where parameter a* takes positive values for reddish colors and negative values for the green ones, parameter b* takes positive values for yellowish colors. Negative values for the bluish ones, and parameter L* is an approximate luminosity measurement. Luminosity is the property according to which each color can be considered as an equivalent to a member of a grayscale between black and white. The total color difference (DE) was calculated using the following formula [29]:(1)$$\Delta E=\sqrt{{\left(\Delta L^{*}\right)}^{2}+{\left(\Delta a^{*}\right)}^{2}+{\left(\Delta b^{*}\right)}^{2}}$$

The following ΔE values are valid universally:A normally invisible difference1–2 Very small difference, only obvious to a trained eye2–3.5 Medium difference, also obvious to an untrained eye3.5–5 An obvious difference>5 A very obvious difference

Non-irradiated DC and AC were characterized by the L* > 92 parameter value and low values of the a* and b* parameters. This arrangement of parameters corresponds to pure white. After irradiation with a dose of 50 kGy, a color change to cream-like was noticed for both tested compounds, which is related to a decrease in the value of the L* parameter and an increase in the values of a* and b* parameters. As the dose increased, the color deepened and the value of the L* component decreased, and the a* and b* values increased. The calculated values of the ΔE for the dose of 50 kGy were 4.65 for AC and 7.42 for DC, and it increased with subsequent doses, reaching the highest values for the samples irradiated with the dose of 400 kGy (19.78 for DC and 14.39 for AC). The ΔE color discrimination index according to the CIEL*a*b* system confirms that the samples irradiated with a dose >25 kGy give a clear color difference in relation to the non-irradiated samples (Figure 2). The same result was obtained in the organoleptic analysis, where the color change was noticed with visual inspection (naked eye).

In most of the described drugs, color changes were observed after irradiation, which may be caused by the formation of colored products of radiolysis or by defects in the crystal lattice and free radicals trapped in them. It is related to the formation of the so-called color centers (i.e., defects in the crystal lattice causing the absorption of visible light at fixed frequencies, which in turn leads to the color of the crystals). The melting point determined by the capillary method was 153.0 °C for AC and 291.0 °C for DC, while after irradiation with a dose of 25 kGy it decreased to 290.0 °C only in the case of DC. It is further reduced by increasing the absorbed dose to 286.0 °C after irradiation with a dose of 400 kGy. On the other hand, for AC the melting point decreased only after the application of the 100 kGy dose (Table 1). When analyzing the DSC curves, it was noticed that the melting point of both compounds decreased only after the irradiation of the dose of 400 kGy by 9.0 °C and 1.0 °C for DC and AC, respectively (Table 1). In the microscope examination using magnification from 10 to 40 times, single crystals were observed before and after irradiation (Figure 3).

During preliminary measurements of the EPR spectrum, a wide and narrow component with significantly different spectral parameters was observed in the exposed samples, therefore the measurements were divided into two ranges. A wide scan of the magnetic field was performed to more precisely characterize the broad component. Then the span was narrowed and a scan was made for the narrow component. The stability of the radicals inducted by the radiation was tested by taking measurements at seven-day intervals. The first measurements were made about 10 h after irradiation for samples exposed to a dose of 25 kGy. No significant differences in collected lines were observed over the three-week period of time. All of the samples were in the form of powders. Relatively small power was used to collect spectra to avoid saturation effect. Typically microwave power was set to circa 0.3 mW. After recording, the spectra were corrected according to the art by removing the baseline and de-noising with a low-pass filter and smoothing. The first analysis was performed on the derivative of the absorption curve. Since relative line intensity is influenced by the quality factor of the cavity line, intensities of the samples were compared with the line intensity of the reference, which was assembled in the resonator during the experiments. After the double integration of the first derivative curve the intensity proportional to the number of spins per 1g mass of each sample was obtained. Low range experiment. The EPR spectra parameters were collected in Table 2. The spectrometer settings: modulation frequency 10 kHz, center field 338 mT, sweep width 15 mT, sweep time 480, modulation amplitude 800 uT, microwave attenuation 25 dB, RF frequency 9.457177 GHz, temperature in the cavity 28 °C. The narrow components for the AC 25 kGy sample were characterized by the highest intensity per mass of the sample and two components with different relaxation times were registered. It is worth mentioning that the substances that were not exposed to AC and DC irradiation showed no measurable number of radicals. That is clear evidence that the high dose rate of radiation can permanently damage the internal structure of substances. The results are shown in Figure 4. Extended range experiment. The EPR spectra parameters were collected in Table 3 and shown in Figure 5. Spectrometer settings: modulation frequency 10 kHz, center field 336 mT, sweep width 500 mT, sweep time 480, modulation amplitude 700 uT, microwave attenuation 25 dB, RF frequency 9.457177 GHz, temperature in the cavity 28 °C.

Radicals induced by ionizing radiation for DC 25 kGy show the highest intensity and at the same time they are characterized by about twice shorter relaxation time than for other substances. A significantly different gyromagnetic coefficient was also observed, which amounted to g = 2.08. As expected [30,31,32], the high dose of irradiation creates the numerous free radicals in the studied samples. Exponential decay of unstable free radicals was expected to be seen in registered data characterized by relatively short life time (in hours) such as, for example, what was described in the case of doxorubicin and epirubicin. However, in the analyzed samples only signals from stable radicals were registered. The highest concentration of free radicals inducted by the radiation in total (summed low and extended range) was observed in DC 25 kGy. Samples without exposure to radiation contain almost no free radicals. For AC 25 kGy the stable radicals concentrations per one gram of mass increased more than three times. The initial concentration of radicals in the AC sample may be due to aging processes or synthetic impurities (substrates, catalysts), the subsequent exposure to ionizing radiation caused additional damage and the lack of possibility of recombination (Figure 4d). The initial concentration of radicals in the AC sample may be due to aging processes or synthetic impurities (substrates, catalysts).

The pharmacopeia does not require research on the presence of free radicals in irradiated drugs, however, the knocking out of an electron from a molecule is usually the first effect of ionizing radiation, therefore, this method is very often used to assess the effect of radiation on drugs. Unfortunately, there are also no standards for the content of free radicals in a unit of mass or volume of a drug, so this analysis does not provide an unambiguous answer to the question of whether the compound can be sterilized by radiation, but by providing information on the number of free radicals, their structure and lifetime, it can significantly facilitate establishing the mechanism of the radio degradation reaction. The effect of radiation depends on the structure of a particular compound. For this purpose, FT-IR and ^1^H-NMR analyses were performed, which allowed determining changes in the physical properties of substances after irradiation. FT-IR spectroscopy identifies chemical bonds in molecules, determines the functional groups, and confirms the changes after irradiation. The FT-IR spectra for DC and AC before irradiation were consistent with the literature standard substances [33], and after irradiation with a dose of 400 kGy (16 times the standard dose), no qualitative changes were observed (i.e., no new bands appeared, and none of the existing bands disappeared). The only observed difference was the decrease in transmittance values observed for AC irradiated with a dose of 200 and 400 kGy, especially in the range from 1000 to 2000 cm^−1^, in the area of the occurrence of C=O and C-O stretching bonds characteristic for the ester group (Figure 6), which may indicate the destructive effect of ionizing radiation in the chemical structure of AC. The purity graph shows the correlation curve of the spectral intensity of the irradiated compounds with respect to the spectra of non-irradiated compounds. We obtained a high purity factor for DC, indicating high spectral matching, while for AC this coefficient decreased with increasing the absorbed dose (Table 4, Figure 7) [34].

The solid-state ^1^H NMR studies were conducted to confirm the effect of irradiation on the change of molecular structure and dynamics of the investigated drugs. The spin-lattice relaxation times T1 were determined using the standard saturation recovery sequence ending with a solid echo. The time dependence of the magnetization M(t) was fitted to a single exponential function M(t) = M0 (1 − exp(−t/T1) to within experimental error, where M0 was the equilibrium value, and T1 was the spin-lattice relaxation time. The uncertainties were approximately ± 10% for all measurements. The temperature dependence of the spin-lattice relaxation times T1 for aceclofenac (AC) and diclofenac (DC) was shown in Figure 8.

For AC a broad, slightly asymmetric and shallow minimum was observed at the temperature of 145 K. The value of T1 recorded at the minimum was 58 s. For irradiated samples, the minimum associated with this motion is also visible but the depth of the observed minimum decreases with the increase of the irradiation dose. This means that radiation causes dissociation of the ester group. For AC irradiated with a dose of 400 kGy, the course of T1 relaxation times as a function of temperature is very similar to the course of relaxation times for DC. For DC, no minimum appears at the temperature curve of the relaxation times confirming a lack of R = –CH_2_–COOH group. Moreover, the times T1 decrease slightly from the value 348 s at temperature 80 K to 76 s at 345 K. A very long relaxation time for parent drugs indicates a hindered molecular dynamic in these systems. It was well documented in literature [35] that COOH groups in AC and DC show a high ability to form hydrogen bonds, which stabilizes and stiffens the structure. The minimum of the relaxation times for AC, taking into account the differences in the structure of both drugs, was attributed to the hindered rotation of the R = –CH_2_–COOH ester group in AC. For irradiated samples, the minimum associated with this motion is also visible but the depth of the observed minimum decreases with the increase of the irradiation dose. It was assumed that radiation may cause dissociation of ester groups. For AC irradiated with a dose of 400 kGy, the course of T1 relaxation times as a function of temperature is very similar to the course of relaxation times for DC. It is known from the literature that AC easily degrades into diclofenac [36]. In addition, shorter relaxation times are clear proof of the presence of free radicals in the samples, which are responsible for shortening the relaxation times and the lack of temperature dependence of the relaxation times. The NMR data were interpreted using the dipolar theory described by Bloembergen, Purcell and Pound [37]. The relaxation times T1 due to the 1H-1H spin-spin interaction modulated by internal motion that may be occurring on the NMR time scale [38]. The following reorientations for AC were assumed: in the low temperature below 80 K, the proton jump between two positions in the hydrogen bond, the reorientation of the ester group, and at the high temperature minimum connected with the reorientation of the whole molecule in the melting range of temperature. The relaxation rate of the multi-proton system can be given by a Woessner formula [39]:(2)$$\frac{1}{T_{1}}={\left(\frac{1}{T_{1}}\right)}^{\mathrm{meting}}+{\left(\frac{1}{T_{1}}\right)}^{\mathrm{ester}\;\mathrm{group}}+{\left(\frac{1}{T_{1}}\right)}^{\mathrm{proton}\;\mathrm{jump}\;}$$
where each contribution is expressed by following Equation (3) [40,41]:(3)$${\left(\frac{1}{T_{1}}\right)}^{*}=\frac{2}{3}\gamma^{2}\Delta M_{2}\left(\frac{\tau_{c}}{1+\omega_{0}^{2}\tau_{c}^{2}}+\frac{4\tau_{c}}{1+4\omega_{0}^{2}\tau_{c}^{2}}\right)$$
where: *** concerns the contribution of particular reorientation, γ is a gyromagnetic ratio of protons, ΔM_2_ is a reduction of the second moment. The correlation time is assumed to be given by the Arrhenius formula: for energy activation, Ea is a pre-exponential factor and R is the universal gas constant. Table 5 collects the activation parameters obtained by numerical fit of Equations (2) and (3) to the experimental relaxation times in the whole temperature range for parent and irritation of AC.

The reorientation of the ester group for the AC is described by activation energy E_a_ = 7.0 kJ/mol and the correlation time τ_o_ = 81.9 × 10^−11^ s. For the motion of the whole molecule, the following activation parameters are estimated: activation energy E_a_ = 10.6 kJ/mol, correlation time τ_o_ = 2.2 × 10^−7^ s. The third low temperature motion described by the relaxation time $\tau_{0}$= 2.$8\times 10^{-11}$ s and low activation energy $E_{a}$= 0.5 (kJ/mol) could be related to the jumps of the proton atom in hydrogen bonds. Regarding obtained activation parameters the effect of radiation is strongly visible for a dose of 400 kGy. For AC 400 kGy, the energies of local motions are smaller than pure AC and the drug irradiated with a dose of 25 kGy, while the correlation times of these movements are longer, which indicates a greater disorder of the system (reduction of the degree of crystallinity of the system). Summing up, it can be concluded that the dose of 25 kGy causes little changes in the physicochemical properties of the drug, in contrast to the dose of 400 kGy, which causes decomposition of the drug. The HPLC method was used to determine the content of DC and AC before and after irradiation, which was validated according to the ICH guidelines [27]. In the chromatogram of the AC before irradiation, one peak with a retention time of 12.26 min, while for DC two signals with retention times of 1.75 min and 11.46 min were observed. After irradiation with the dose from 25 to 200 kGy, no additional signals were observed in the chromatograms of both substances. Only after irradiation with the dose of 400 kGy, additional peaks appeared for DC at t_R_ = 9.82 min, and for AC at t_R_ = 1.75 min. Within the concentration range from 0.5038 × 10^−4^ to 6.0456 × 10^−4^ g/mL for AC and 0.5014 × 10^−4^ to 6.0168 × 10^−4^ g/mL for DC the method was linear, characterized by a very good correlation coefficient (r = 0.9975–0.9978) and high precision (RSD = 1.07–1.15%). The method was validated according to the Guidelines of the International Conference on Harmonization [27]. The content of DC and AC was calculated by reference to the standard. The difference in the content between the irradiated DC and AC and the calculated content of DC and AC brought information on the loss of the active substance as a result of irradiation. Table 6 shows the results of DC and AC quantification before and after irradiation. The decrease in the content after irradiation with the dose of 400 kGy is slight and amounts to 1.84% and 1.98% for DC and AC, respectively.

## 4. Conclusions

The results presented above confirmed that the physicochemical properties of DC and AC did not change under the influence during electron beam irradiation with the standard dose of 25 kGy, which indicates the possibility of sterilization by this method. Although free radicals are formed during irradiation, there is a slight color change and a change in melting point determined by the capillary method. The color change of irradiated samples may be related to the appearance of free radicals detected by the EPR, but may also be due to the appearance of trace amounts of radiolysis products. No decrease in the content determined by the HPLC method was observed. Only after irradiation with a dose of 400 kGy were changes in the molecular structure observed (especially in the case of AC). The FT-IR spectrum showed a decrease in transmittance at the wavenumbers corresponding to the ester group (ν = 1770, 1717 cm^−1^), which was also confirmed by the ^1^H-NMR method.

## Figures and Tables

**Figure 1 pharmaceutics-14-01331-f001:**
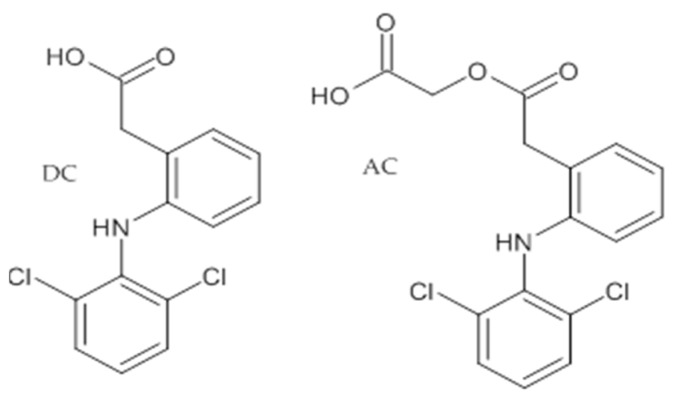
Structure of dicklofenac (DC) and aceclofenac (AC).

**Figure 2 pharmaceutics-14-01331-f002:**
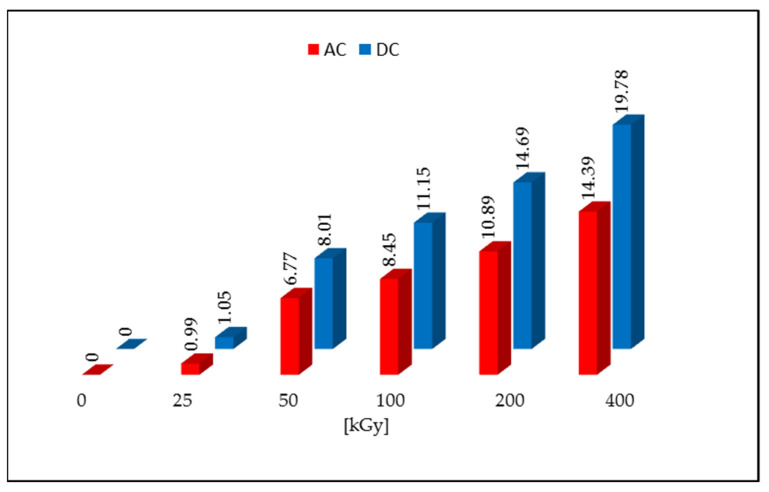
Color changes after irradiation with an electron beam.

**Figure 3 pharmaceutics-14-01331-f003:**
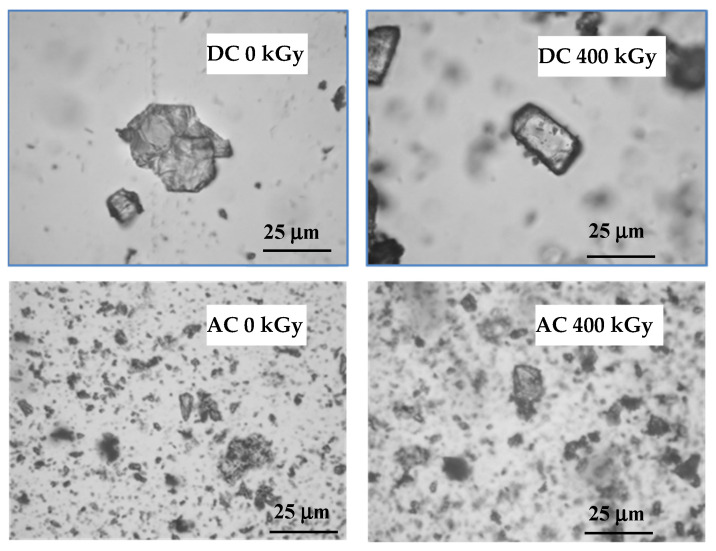
Optical microscope photographs of test compounds before and after irradiation.

**Figure 4 pharmaceutics-14-01331-f004:**
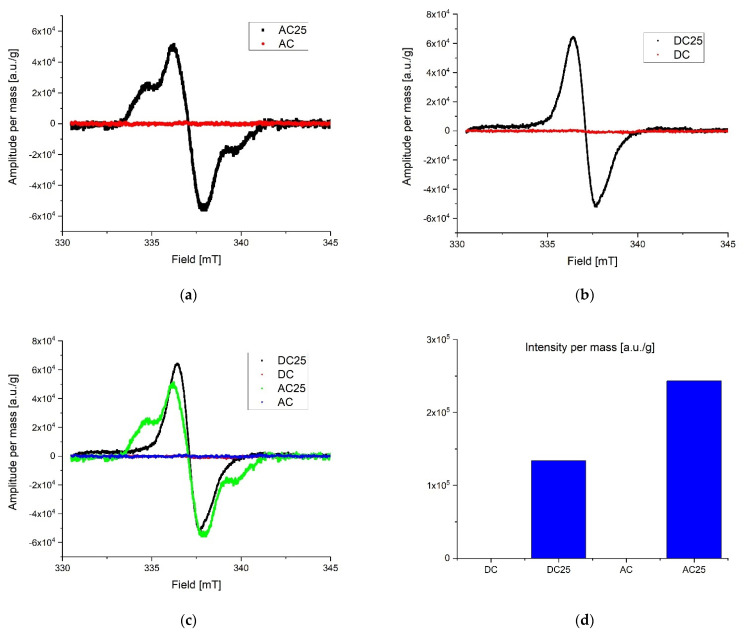
Low range EPR spectra: (**a**) overlap AC and AC 25; (**b**) overlap DC and DC 25; (**c**) overlap AC, AC 25, DC, DC 25; (**d**) intensity proportional to the number of spins in arbitrary units per 1 g mass.

**Figure 5 pharmaceutics-14-01331-f005:**
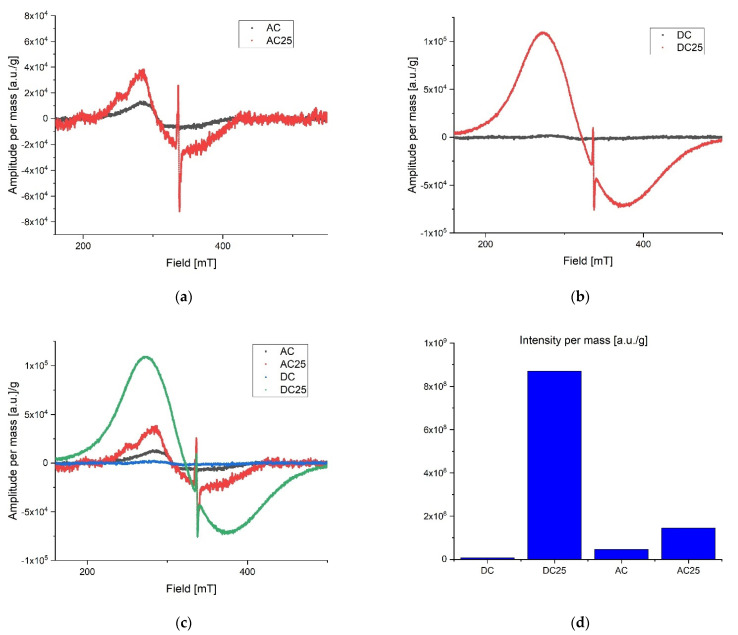
Extended range EPR spectra: (**a**) overlap AC and AC25; (**b**) overlap DC and DC25; (**c**) overlap AC, AC25, DC, DC25; (**d**) intensity proportional to the number of spins in arbitrary units per 1g mass.

**Figure 6 pharmaceutics-14-01331-f006:**
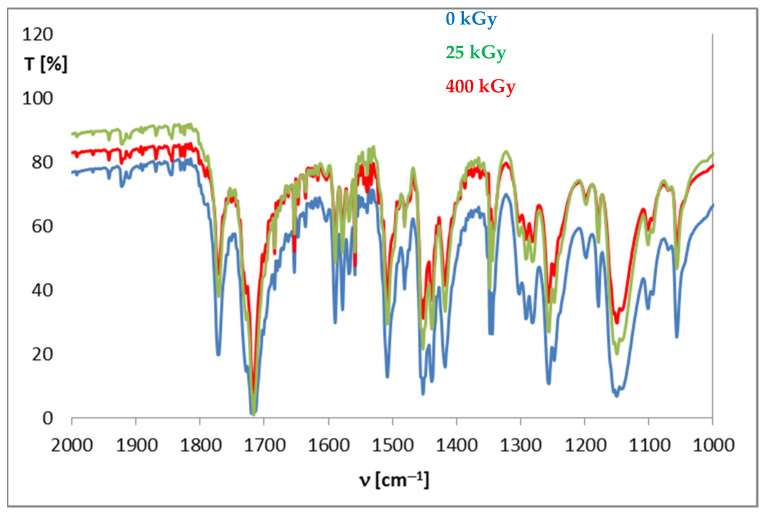
FT-IR spectra for aceclofenac before and after irradiation.

**Figure 7 pharmaceutics-14-01331-f007:**
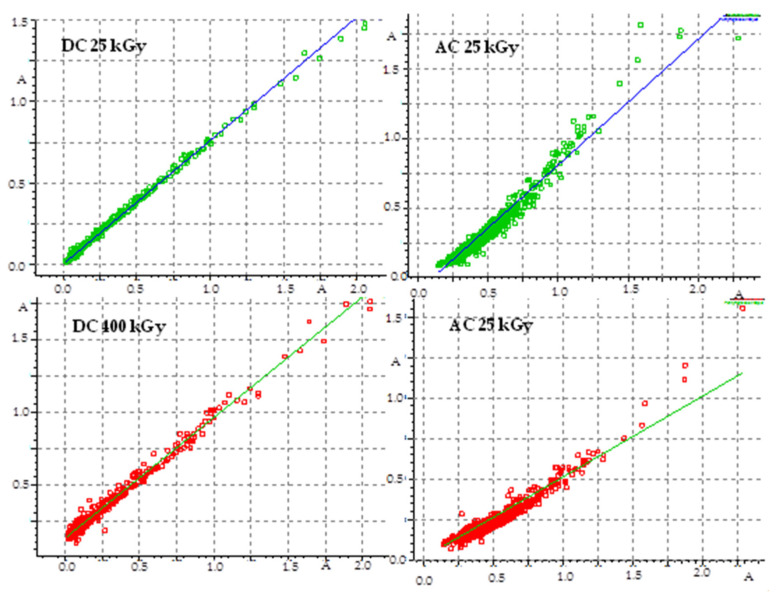
Graph of FT-IR spectra compatibility before and after irradiation.

**Figure 8 pharmaceutics-14-01331-f008:**
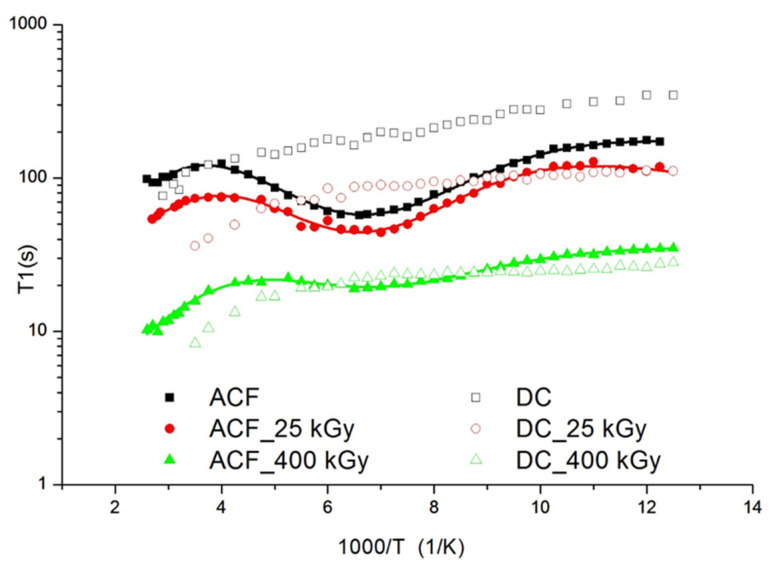
Temperature dependence of the spin-relaxation time T1 in the laboratory frame for Aceclofac (full) and Diclofenac (open) drugs. The solid line represents the best theoretical fit of Equations (2) and (3) to the experimental data.

**Table 1 pharmaceutics-14-01331-t001:** Melting point results.

Dose [kGy]	Melting Point [°C]			
	Capillary Method		DSC Method	
	DC	AC	DC	AC
0	291.0	153.0	284.0	155.0
25	290.5	153.0	284.0	155.0
400	286.0	151.0	275.0	154.0

**Table 2 pharmaceutics-14-01331-t002:** Low range EPR spectra parameters.

Parameters	DC 0 kGy	DC 25 kGy	AC 0 kGy	AC 25 kGy
Line width [mT]	−	1.26	−	1.75
Line width 2 [mT]	−	−	−	4.41
g	−	2.001	−	2.0035
Intensity per mass [a.u./g]		133,810		243,214

**Table 3 pharmaceutics-14-01331-t003:** Extended rangr EPR spectra parameters.

Parameters	DC 0 kGy	DC 25 kGy	AC 0 kGy	AC 25 kGy
Line width [mT]	42.821	101.229	40.3642	43.3694
g	2.208	2.08	2.2054	2.2131
Intensity [a.u./g]	7,846,892	869,600,000	45,774,601	144,154,914

**Table 4 pharmaceutics-14-01331-t004:** Purity factor for DC and AC in FT-IR method.

Dose [kGy]	25		50		100		200		400	
	DC	AC	DC	AC	DC	AC	DC	AC	DC	AC
Conformity factor	0.9957	0.9759	0.9826	0.9596	0.9776	0.9643	0.9874	0.9693	0.9809	0.9471

**Table 5 pharmaceutics-14-01331-t005:** Activation parameters of assumed motions obtained for all samples of AC. The values of uncertainty of the estimated parameters were lower than 10%.

Sample	Reorientation of the Whole Molecule		Hindered Rotation of-CH_2_COOH Group		Jump of Proton in Hydrogen Bonds	
	τ_0_ (s)	E_a_ (kJ/mol)	τ_0_ (s)	E_a_ (kJ/mol)	τ_0_ (s)	E_a_ (kJ/mol)
AC	2.2 × 10^−7^	10.6	1.9 × 10^−11^	7.0	2.8 × 10^−11^	0.5
AC_25 kGy	3.5 × 10^−7^	6.0	2.1 × 10^−11^	6.7	2.4 × 10^−11^	0.5
AC 400 kGy	2.02 × 10^−10^	11.0	6.3 × 10^−11^	5.2	9.3 × 10^−11^	0.2

**Table 6 pharmaceutics-14-01331-t006:** HPLC analysis of AC and DC.

Dose [kGy]	Content [%]	
	DC	AC
25	98.64	100.52
50	98.65	100.44
100	98.67	99.84
200	98.57	99.23
400	96.86	98.02

## Data Availability

The data presented in this study are available in the article.

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
