# Peer review of "Electron Beam Radiation as a Safe Method for the Sterilization of Aceclofenac and Diclofenac—The Usefulness of EPR and 1H-NMR Methods in Determination of Molecular Structure and Dynamics"

_pharmaceutics, 2022, doi:10.3390/pharmaceutics14071331_

Round 1
Reviewer 1 Report
In this article the authors evaluated the radiochemical stability of Diclofenac and Aceclofenac when exposed to radiation in the form of an electron beam. The analyzes are complete but the aim of the work is not clear.
What is the innovation of this project? Other sterilization methods are already reported in the literature, what is the advantage of this method?
Abstract is not sufficiently informative. Authors should add an introduction to explain the purpose of the work and reduce some very detailed results, such as the melting point.
"Aceclofenak" should be corrected (keywords and line 87) with "Aceclofenac".
In paragraph 2.1 the numbers of the molecular formulas should be written in subscript (lines 84 and 87).
In lines 128-129, 165, 345-346 the commas should be corrected with points.
Author Response
Response to Reviewer 1 Comments
We wish to thank the Reviewer for carefully prepared reports on our paper “Electron beam radiation as a safe method for the sterilization of aceclofenac and diclofenac. The usefulness of EPR and 1H-NMR methods in determination of molecular structure and dynamics ”. Please find enclosed a new version of this paper rewritten taking into regard all the comments and suggestions. English native speaker carefully revisited this manuscript. Our detailed responses are given below.
- An explanation of the aim of the study was added to the introduction section.
- Keywords and line 87 were corrected.
- Line 84 and 87 In paragraph 2.1 the numbers of the molecular formulas were written in subscript
- Line 128-129, 165, 345-356 the commas should be corrected with points

Reviewer 2 Report
Pharmaceutics 2022, 14,
The paper describes analysis of results of irradiation treatment of two pharmaceuticals. The sterilization dose is generally 25 kGy, but in order to see real radiation effects they increased the dose in their investigations up to 400 kGy.
Certainly, the paper needs a rather careful revision.
1. In connection with the structure of the manuscript I will make three remarks. The Introduction is too long; it mainly describes what the selected medicines are used for. The Materials and Methods section is too much detailed. There is no real evaluation or discussion section. Here the Conclusion part tries to fulfill this need.
2. There are two serious terminology problems in the manuscript. The authors often use the terms radiation dose or irradiation dose. The correct term is absorbed dose (or simply dose). The authors often use the term radiochemistry (e.g., radiochemical stability). Radiochemistry deals with the application of radioactive isotopes, e.g., tracer technique. Here the authors used radiation chemistry, which is concerned with the effects of ionizing radiation.
3. I think there is some discrepancy with the concentration ranges. E.g., in Line 164 we read: The concentrations ranged from 164 0,5038 x 10-4 to 6,0456 x 10-4 mg/mL and from 0,5014 x 10-4 to 6,0168 x 10-4 for AC and DC, respectively. The unit is not given for the second compound, at other places in the manuscript it is given in g/ml. I think for the first compound also this unit, g/ml and not mg/ml should be applied. Please, check everywhere in the manuscript.
4. Between the unit and the number space is required, e.g., Line 128 5mm. Please check everywhere.
5. During irradiation the temperature of the sample increases. What was the raise of the temperature? At the higher doses, the observed changes are not due to temperature effects.
6. Line 30. Determination of molecular structure and dynamics of DC and AC after irradiation was possible by using EPR and 1H-NMR 1H-NMR methods. After irradiation the molecular structure and dynamics (?) of DC and AC were determined by using EPR and 1H-NMR 1H-NMR methods.
7. What was the accuracy of melting point determination? Line 208: 290.9 oC and 290.5 oC differ really?
8. Line 105. auto-calibration by white and black by white and black colors
9. Line 260. Sample without exposing on radiation contains almost non free radicals. English!
10. Line 261. The initial concentration of radicals in the AC sample may be caused by the aging processes taking place in the substrates or beginning impurities Funny!
11. Fig. 7 needs explanation!
12. Line 313. which are responsible for shortening the relaxation times, and no dependence of relaxation times on temperature. English!
13. Table 6. I do not understand. The content in percentage decreases with the absorbed dose, in the changes of concentrations there is no definite trend.
14. Line 365. influence of ionizing radiation in the form of an electron beam in the standard dose of 25 kGy influence during electron beam irradiation with the standard dose of 25 kGy
15. Line 366. formed after irradiation formed during irradiation
A very careful revision is needed!
Author Response
Response to Reviewer 2 Comments
We wish to thank the Reviewer for carefully prepared reports on our paper “Electron beam radiation as a safe method for the sterilization of aceclofenac and diclofenac. The usefulness of EPR and 1H-NMR methods in determination of molecular structure and dynamics ”. Please find enclosed a new version of this paper rewritten taking into regard all the comments and suggestions. English native speaker carefully revisited this manuscript. Our detailed responses are given below.
- The Reviewer suggests that the introduction is too extensive. We believe that this section is a good introduction to the application of the radiation sterilization method for pharmaceutical forms requiring sterility.
- The reviewer suggests that The Materials and Methods section is too much detailed.
The methodology and materials have been described in detail so that the experiment can be replicated
- As suggested by the Reviewer, throughout the manuscript, we used the terminology absorbed dose instead of radiation dose
- As suggested by the Reviewer, throughout the manuscript, we used the terminology radiation instead of radiochemistry
- As suggested by the Reviewer, the concentration in g/mL was introduced throughout the study
- Lines 128 added a 5 mm spacing
- Irradiation was performed according to the procedure at room temperature
- Line 30: “Determination of molecular structure and dynamics of DC and AC after irradiation was possible by using EPR and 1H-NMR 1H-NMR methods” has been replaced with: "The influence of irradiation on changes in molecular dynamics and structure has been observed by 1H-NMR and EPR studies"
- Line 208 Melting point values ​​have been rounded to the nearest whole value
- Line 105 was changed as suggested by the Reviewer
- Line 261 Non-irradiated samples contain trace amounts of free radicals as can be seen in Fig 4d
The sentence “The initial concentration of radicals in the AC sample may be caused by the aging processes taking place in the substrates or beginning impurities” has been replaced with “The initial concentration of radicals in the AC sample may be due to aging processes or synthetic impurities (substrates, catalysts) . "
- 7. In FT-IR spectra made for AC and DC before and after irradiation with doses ranging from 25 to 400 kGy, all bands corresponding to the most important elements of the chemical structure of both compounds were observed at the appropriate wave numbers, which may indicate that ionizing radiation in the form of an electron beam did not change their chemical structure.
In order not to increase the volume of the work, the reference [34] has been added, in which the formulas for calculating the conformity factor are described and explained.
- In line 313, the sentence: "which are responsible for shortening the relaxation times, and no dependence of relaxation times on temperature" has been replaced with: "which are responsible for shortening the relaxation times and the lack of temperature dependence of the relaxation times."
- Table 6 "Concentration" column has been removed
- Line 365 - changed as suggested by the Reviewer
- Line 366 - changed as suggested by the Reviewer

Reviewer 3 Report
This study presents the response of two compounds to radiation using an electron beam and indicating that it is a safe method for sterilization. A series of studies are presented in order to know the impact of this radiation. In the summary, the relevance of the study can be improved, which will allow to identify the impact and novelty of the proposed study. Which is recommended and would cause a greater impact.
The state of the art is adequately preying, identifying the various studies carried out in the fear of interest. Which consists of determining the radiochemical stability of the CC and AC using spectrophotometric, thermal and chromatographic methods.
Some of the methodologies would be advisable to support with literature. They are described in a timely manner. And such support would support the application of certain conditions of the study.
More information about the change of color when applying radiation, to which these changes are attributed according to the literature.
Figure 3, show scale if possible.
Broaden discussion regarding the presence or absence of free radicals, with respect to radiation exposure. This in order to expand the understanding of the associated phenomenon. Which is of relevance to the study
Line 269-272: "The effect of radiation depends on structure of particular compound. For this purpose, FT-IR and 1H-NMR analyses were performed, which allowed determining changes in the physical properties of substances after irradiation....." It is recommended to expand analysis with the support of specialized literature. As FT-I and NMR determine changes in physical properties, based on which couplings or presence of functional groups, is related to such physical change. Is there any evidence, for example a scanning electron microscopy (SEM) study?
The conclusions can be improved by presenting the conclusion of why the change in coloration and for the case of change in structure. Which would complement what was indicated in that point.
Author Response
Response to Reviewer 3 Comments
We wish to thank the Reviewer for carefully prepared reports on our paper “Electron beam radiation as a safe method for the sterilization of aceclofenac and diclofenac. The usefulness of EPR and 1H-NMR methods in determination of molecular structure and dynamics ”. Please find enclosed a new version of this paper rewritten taking into regard all the comments and suggestions. English native speaker carefully revisited this manuscript. Our detailed responses are given below.
- Explanation of the appearance of the color in irradiated samples:
In most of the described drugs, color changes were observed after irradiation, which may be caused by the formation of colored products of radiolysis or by defects in the crystal lattice and free radicals trapped in them. It is related to the formation of the so-called color centers, i.e., defects in the crystal lattice causing the absorption of visible light at fixed frequencies, which in turn leads to the color of the crystals.
The mentioned paragraph was provided in the description of Fig. 2.
2. As the reviewer suggested, the scale in Fig. 3 was added, and the caption under Figure was additionally extended.
Before: Photographs of tested compounds before and after irradiation
Now: Optical microscope photographs of test compounds before and after irradiation.
3. Extended discussion on the appearance of free radicals in irradiated drugs. The paragraph below was provided in the description of Fig. 5
The pharmacopeia does not require research on the presence of free radicals in irradiated drugs, however, the knocking out of an electron from a molecule is usually the first effect of ionizing radiation, therefore, this method is very often used to assess the effect of radiation on drugs. Unfortunately, there are also no standards for the content of free radicals in a unit of mass or volume of a drug, so EPR method does not provide an unambiguous answer to the question of whether the compound can be sterilized by radiation, but by providing information on the number of free radicals, their structure and lifetime, it can significantly facilitate establishing the mechanism of the radiolysis reaction.
4. Line 269-272:
We would like to thank the Reviewer for his valuable tips on the possibility of using other research techniques (SEM), which will be used in further research on the influence of ionizing radiation on medicinal substances. In work, we used an optical microscope for preliminary examinations (Fig. 3).
5. The conclusions section was supplemented with the following sentence:
The color change of irradiated samples may be related to the appearance of free radicals detected by the EPR, but may also be due to the appearance of trace amounts of radiolysis products.
